# Successful Integration of Clinical Pharmacists in an OPAT Program: A Real-Life Multidisciplinary Circuit

**DOI:** 10.3390/antibiotics11081124

**Published:** 2022-08-19

**Authors:** Sara Ortonobes, Abel Mujal-Martínez, María de Castro Julve, Alba González-Sánchez, Rafael Jiménez-Pérez, Manuel Hernández-Ávila, Natalia De Alfonso, Ingrid Maye-Pérez, Teresa Valle-Delmás, Alba Rodríguez-Sánchez, Jessica Pino-García, Mònica Gómez-Valent

**Affiliations:** 1Pharmacy Department, Consorci Corporació Sanitària Parc Taulí, Universitat Autònoma de Barcelona, 08208 Sabadell, Barcelona, Spain; 2Parc Taulí Research and Innovation Institute Foundation (I3PT), 08028 Sabadell, Barcelona, Spain; 3Hospital at Home Unit, Consorci Corporació Sanitària Parc Taulí, Universitat Autònoma de Barcelona, 08208 Sabadell, Barcelona, Spain

**Keywords:** OPAT, multidisciplinary circuit, clinical pharmacist

## Abstract

Outpatient parenteral antimicrobial therapy (OPAT) programs encompass a range of healthcare processes aiming to treat infections at home, with the preferential use of the intravenous route. Although several barriers arise during the implementation of OPAT circuits, recent cumulative data have supported the effectiveness of these programs, demonstrating their application in a safe and cost-effective manner. Given that OPAT is evolving towards treating patients with higher complexity, a multidisciplinary team including physicians, pharmacists, and nursing staff should lead the program. The professionals involved require previous experience in infectious diseases treatment as well as in outpatient healthcare and self-administration. As we describe here, clinical pharmacists exert a key role in OPAT multidisciplinary teams. Their intervention is essential to optimize antimicrobial prescriptions through their participation in stewardship programs as well as to closely follow patients from a pharmacotherapeutic perspective. Moreover, pharmacists provide specialized counseling on antimicrobial treatment technical compounding. In fact, OPAT elaboration in sterile environments and pharmacy department clean rooms increases OPAT stability and safety, enhancing the quality of the program. In summary, building multidisciplinary teams with the involvement of clinical pharmacists improves the management of home-treated infections, promoting a safe self-administration and increasing OPAT patients’ quality of life.

## 1. Introduction

According to the definition proposed by Tice et al., OPAT (outpatient parenteral antimicrobial therapy) refers to the provision of parenteral antimicrobial therapy in at least two doses on different days without intervening hospitalization [1]. It is, therefore, a broad concept that encompasses not only the type of antimicrobial (antibacterial, antifungal, or antiviral) but also different routes and modes of administration. Specifically, this procedure can be applied, for example, in patients’ homes, primary care centers, day hospitals, or emergency rooms. All these possibilities have given rise to the existence of different terms for its denomination, but the most widely used and widespread term in English is the acronym OPAT. However, another existing term is HIVAT [2,3] (home intravenous antimicrobial treatment), which although less used, better defines the care model in Spain because it refers to treating the infection at the patient’s home and, moreover, it also highlights the preferred use of the intravenous route for antimicrobial administration.

In Spain, OPAT is almost exclusively carried out by the framework of hospital-at-home (HaH) units [4]. HaH is a model of care organization that provides specialized care for highly complex patients who, due to their clinical stability and favorable sociocultural environment, can be safely treated at home [5]. HaH units emerge as an effective alternative to hospital admission, capable of providing similar levels of diagnostic resolution, medical care, and pharmacological treatment as those offered in conventional hospitalization [6]. Remarkably, advances in surgery (minimally invasive procedures), pharmacology (with the emergence of new antimicrobials), and telemedicine (improving communication between patients and health services and the miniaturization of infusion devices) have allowed the expansion of HaH units in most countries [7].

Therefore, we understand OPAT as the healthcare service in which the hospital travels to the patient’s home to treat an infectious process using intravenous antimicrobial agents and performing full care for patients who otherwise would have to stay in hospital. The Spanish Society of Hospital at Home (SEHAD) has drawn up a decalogue on OPAT [8] that defines the characteristics, objectives, and needs of this care model, expanding the concept to “Complex OPAT” (Table 1).

Seaton et al. analyzed the implications that the OVIVA (oral versus intravenous antibiotics) randomized study about bone and joint infections could have on OPAT. This study challenged the practice of prolonged intravenous therapy since the non-inferiority of oral antibiotic therapy was demonstrated. The authors state that oral ambulatory antibiotic therapy for these types of infections should be considered as complex outpatient antibiotic therapy [9]. In our opinion, and in agreement with the SEHAD decalogue [8], complex OPAT should have a much broader definition and include intravenous antibiotic treatments performed at patients’ homes. We believe that defining an OPAT treatment as complex should refer to a high complexity in technical preparation, either due to the need to treat severe infections or infections caused by difficult-to-treat microorganisms. Multidrug-resistant infections generally require administering antibiotics in several doses per day as well as to administer combinations of different antimicrobials. In these cases, it is essential to achieve an optimal and safe self-administration of the treatment by the patient (or a caregiver). Due to the complexity of the whole OPAT program, it should be closely monitored by a multidisciplinary team [10].

### Main Barriers for Efficient OPAT Implementation

As occurs with in-hospital patients, the inappropriately prolonged use of antimicrobials in an OPAT setting can induce the appearance of bacterial resistance. In both cases, this results in increased morbidity, hospitalization rates, economic costs, and mortality [11]. However, it must be taken into account that the OPAT scenario presents some particular issues compared to in-hospital therapy (e.g., pharmacokinetic and pharmacodynamics (pK/pD) requirements to determine the infusion method or the selection of appropriate venous catheters) [12,13]. Therefore, the involvement of a multidisciplinary team is essential to achieve successful results in OPAT [14], overcoming the potential barriers arising throughout the process.

In this sense, Hamad et al. identified the main barriers for safe OPAT care [15]. The most challenging barriers described were: (i) a lack of immediateness in obtaining laboratory results, (ii) low funding for OPAT implementation from healthcare institutions, (iii) weak communication with other OPAT service providers, (iv) a failure in patients’ follow-up by infectious disease service, (v) a lack of laboratory result checks during OPAT administration, and (vi) difficulties in tracking OPAT patients through the medical record system.

Moreover, the implementation of an OPAT circuit involves a higher workload for nurses and caregivers, increasing the required time for optimal patient care. It may also be considered a time-consuming circuit, enlarging the response time in the case of complications and increasing the economic costs for the implementation of OPAT devices. In this sense, previous cost-effectiveness studies demonstrating OPAT’s advantages versus in-hospital treatment are available [16]. Another important drawback faced by OPAT implementation is the low funding received from institutional organs, as each healthcare provider struggles to adjust the available resources, resulting in a highly heterogeneous application of OPAT [17] and inefficient communication and coordination between the partners involved. These barriers impair an appropriate control of OPAT safety and may even unnecessarily prolong regimens [18].

In order to solve and avoid all these barriers, the onset of an OPAT multidisciplinary team should consider a series of relevant aspects. The number of professionals involved in an OPAT program has to be adjusted according to the population of a specific area, the required time for assistance in patients’ homes, and the response time required in the case of detection of adverse effects. In addition, all members of the team should possess appropriate expertise in infectious disease management and further skills in intravenous outpatient care. Finally, fast and efficient communication between members of the OPAT team is required in order to share patients’ clinical data, such as laboratory results, in a timely fashion between in-hospital and outpatient services [10,12,19,20,21].

In fact, the wide experience of the healthcare professionals of our HaH unit has allowed them to overcome the described barriers, and nowadays the basic healthcare standards of outpatients are covered to the same extent as in-hospital admitted patients. For example, if required, laboratory analytics can be performed daily, and the results are obtained immediately. Each institution needs to adapt its resources to solve all barriers identified in this section, with the final goal of accomplishing the standards for effectiveness, safety, and cost-effectiveness in OPAT programs.

## 2. Effectiveness, Safety, and Cost-Effectiveness

### 2.1. Clinical Effectiveness of OPAT

Cumulative data in recent years support the effectiveness of OPAT programs. In a systematic review that evaluated the available evidence relating to adult OPAT services, most of the studies presented a successful treatment range above 90%, with a total range of 69–99%. The rate of readmission was heterogeneous, varying between 2 and 26%, although only 4 of 12 studies showed a readmission rate of more than 10%. The most frequent causes of readmission were adverse drug reactions (ADRs), venous access complications and/or infection, or social concerns. The mortality rates were below 1.5% in most of the studies [22].

The variations in readmission rates may be explained by differences in healthcare models, in the methodology used to collect data, and consequently, in data heterogeneity. HaH unit models in Spain report low readmission rates since adverse effects are managed by a physician–nurse team in patients’ homes, thus usually avoiding readmission [23]. This observational study from the Spanish Outpatient Parenteral Antimicrobial Therapy Registry (GESTADE registry) also describes a 91% rate of healing or improvement, leading to successful therapies. It is notorious that, in all cases, patients were contacted daily by telephone and visited at home by nursing staff and physicians, according to an adapted schedule, suggesting that this healthcare model with more intensive care and intervention can be used to treat disease processes of greater complexity and severity [23]. In the same line, further recent studies have shown similar rates of success, reinforcing the idea of OPAT as an effective and safe therapy as well as an alternative to hospitalization [24,25,26].

One of the fundamentals of OPAT model success is the widespread use of elastomeric pump devices. While administration by gravity is still the most widely used, it has been reported that antibiotic concentrations reach appropriate blood levels better by using elastomeric pumps. These data are associated with a higher rate of therapy success using these devices [27]. Thus, the use of elastomeric pumps enables the ambulatory management of patients and ensures easier and more comfortable self-administration.

On the other hand, the type of infection widely varies in the different series studied, complicating the interpretation of the results. Thus, respiratory, urinary, and intra-abdominal infections are the most prevalent in some series [23,24,25], while other studies mostly report osteoarticular and skin and soft tissue infections [28,29,30,31]. Interestingly, the latter type of infections has been associated with a greater treatment failure than other conditions [28]. Likewise, an increased risk of treatment failure and/or relapse seems to be related to a higher Charlson comorbidity score and certain types of infection (mainly bone and joint, respiratory, and intra-abdominal infections and bacteremia without focus) [26,27,28,29,30,31,32].

Other studies analyzed the effectiveness, depending on the antimicrobials used. For instance, Saini et al. showed that monotherapy with ceftriaxone, non-immunosuppressed patients, and an outpatient follow-up within 2 weeks after discharge are related to a significantly reduced risk of 30-day readmission [33]. In parallel, ertapenem OPAT was observed to have excellent effectiveness and safety [25,26], while the effectiveness of other treatments, such as micafungin and further antifungals, is more limited, probably due to their side effects and the greater severity of the infectious process and previous underlying conditions [33,34].

Although there are limited data regarding elderly patients, some studies have evaluated the effectiveness and safety of OPAT in this group of patients. A comparative study between 70-year-old patients and the younger population resulted in no significant differences in treatment success (92% in people over 70 years) or readmission rates (22% in people over 70 years vs. 23% in the younger population) [29]. The same group found that laboratory test parameters and the degree of physical dependence, but not age, were associated with OPAT failure and the readmission rate [30]. Another observational study comparing three groups of patients (<65 years, 66–79 years, and >80 years) showed that there were no differences in hospital readmissions during at-home hospitalization or during the first 30 days after discharge. Older people had similar changes in antibiotic treatment and hospital readmission rates due to poor control of underlying infections but higher readmission rates due to the worsening of underlying diseases than younger adults [35]. Consistently, the analysis of 260 OPAT episodes in the aged population (>75 years) achieved success in 95% of cases, with a median length of 14 days of OPAT, evaluated as safe in 81% of patients [36]. This might suggest that OPAT is also effective in elderly people, showing that this healthcare model has a potential benefit in this population.

Thus, OPAT may be successfully provided in highly complex patients and can be used to treat disease processes of greater complexity, both in elderly comorbid patients and in complex/acute-phase infections.

### 2.2. OPAT Safety

Despite being considered an efficient and safe alternative to prolonged hospital admissions, OPAT programs may involve a series of adverse effects during the treatment, which may lead to OPAT failure (ranging from treatment discontinuation to hospital readmission) [37,38,39]. These collateral effects are mainly related to either venous access (catheter-related complications) or to the antimicrobials themselves (ADRs and drug interactions).

#### 2.2.1. Catheter-Related Complications

Catheter-related complications in OPAT are similar to those appearing in a hospital setting and rarely cause patient readmission [10]. Venous catheter choice in OPAT depends on both the therapy length and the type of antimicrobial used. In general, peripherally inserted central catheters (PICC) and midline catheters are the most common in our OPAT program.

Krein et al. reported that more than 60% of patients with PICC placement describe at least one complication or adverse effect [40]. PICC-related problems include infections, venous thrombosis, and mechanical complications, with global rates of 15.9%, 34.0%, and 40.7%, respectively [41]. A prospective observational study estimated that mechanical issues represent almost half of all PICC-related complications [39]. Further studies described occlusion rates between 2.4% and 6.0% among hospitalized patients [42,43], whereas rates between 4.5% and 7.4% were reported for outpatients [44,45]. An age above 65 years and previous episodes of occlusive events have been identified as risk factors for mechanical complications. Moreover, accidental catheter withdrawal is also common in older patients with PICC placement and becomes challenging for nursing care, especially in patients with cognitive impairment [41].

The estimated rate of bloodstream infections related to PICC is 2.1 per 1000 days of catheter use in hospitalized patients, whereas it is reduced to 1.0/1000 days in outpatients [41]. These differences can be explained because in hospitalized patients PICCs are additionally used for the administration of further therapies beyond antimicrobials (such as parenteral nutrition), which involve higher daily manipulation [41]. In parallel, it was reported that bloodstream-related infections are an extremely rare event in OPAT, with a rate of one event per eight patients followed per year, with local inflammation being the most frequently reported sign [39].

On the other hand, a prospective study in the UK revealed that in OPAT patients catheter-related complications are more common than drug-related adverse effects, as self-administration caused more complications than nursing administration. However, these differences are related to the predominant use of midline catheters in the UK, which are shorter and need to be reviewed more frequently than PICCs [46]. Seo et al. demonstrated that the use of midline catheters for a maximum of 14 days is equally safe for outpatients compared to its use in inpatients [47]. Currently, midline catheters are replacing the use of PICCs in the OPAT program of our HaH unit.

To summarize, the use of intravenous catheters in OPAT programs provides acceptably low rates of infectious, thrombotic, and mechanical complication events, thus demonstrating the safety of these devices in outpatient settings.

#### 2.2.2. ADRs and Drug Interactions

Complications derived from pharmacological treatment are frequent in OPAT programs, and they often involve hospital readmission [10]. In fact, several studies suggest that ADRs occur in 15–30% of patients under OPAT treatment [13]. The chances of developing ADRs are significantly higher in the first two weeks of the treatment, frequently caused by conciliation errors at discharge [48]. In addition, a case rate between 6% and 27% of 30-day readmission due to ADRs is estimated [48,49,50].

The most common ADRs in OPAT patients are cutaneous rashes and gastrointestinal disorders. However, some groups of antimicrobials are related to more specific ADRs. For instance, aminoglycosides are frequently associated with nefro- and ototoxicity, carbapenems are related to an increased risk of convulsions; and beta-lactams can induce hemolytic anemia. To prevent these ADRs, patients’ clinical and analytical parameters should be closely monitored by a multidisciplinary OPAT team at least every 48 h [10].

Next, drug interactions are the second largest group of drug-related complications in OPAT, especially in older patients due to the high polypharmacy degree in this age group [51]. Drug interactions may occur through distinct mechanisms, with enzymatic interaction with CYP450 being the most remarkable. Nevertheless, most of these interactions are easily preventable since they have been widely reported (for instance, the risk of rhabdomyolysis when combining the long-term use of daptomycin with statins) [10].

In this context, the role of the pharmacy department acquires a higher importance to prevent the appearance of ADRs and drug interactions. In fact, they are responsible for checking treatment adequacy, including the appropriateness of the antimicrobial used as well as the treatment regimen, drug stability, mode of administration, and potential interactions with other prescribed drugs [13].

### 2.3. Cost-Effectiveness of OPAT

OPAT has become a well-established and cost-effective measure that leads to the patient being discharged earlier from hospital, improving recovery and avoiding unnecessary hospital stays that might increase the risk of nosocomial infections and bed occupancy. Thus, the OPAT model leads to a more efficient use of healthcare resources and enables higher bed availability. Consequently, OPAT is becoming increasingly widespread around the world thanks to the evident cost-effectiveness that its use entails, including significant savings to national healthcare systems. Nevertheless, pharmacoeconomic analyses comparing OPAT and conventional hospitalization are required, taking into account both direct and indirect costs [52,53,54].

The first cost-effectiveness study was performed by the UK National Health Service and compared the real 2-year expenses of OPAT with the expected derived-costs of a 2-year hospitalization in a conventional hospital and considering the treatment of the same pathology. The study found that OPAT costs only corresponded to 47% of equivalent hospitalization expenses [31]. Although, the extent of savings varies depending on the OPAT healthcare delivery models and the differences between healthcare system financing in each country. In fact, several studies only demonstrate the cost-effectiveness of OPAT by analyzing the annual number of hospitalization days saved [52,53,55]. For instance, the Glasgow group found a savings of 39,035 days in 10 years for 2638 OPAT admissions [52], while the London group found a savings of 7394 days of hospital admission in 44 months for 303 OPAT admissions [53]. However, some studies associating OPAT cost-effectiveness in terms of public healthcare saving can be also found. Along this line, a group in Saudi Arabia reported that switching from inpatient to OPAT care resulted in a cost savings of around USD 4.8 billion in 2.5 years [25].

In Spain, economic studies have also demonstrated the cost-effectiveness of OPAT. An economic analysis carried out in three Spanish HaH units showed that OPAT resulted in lower costs compared to inpatient care [56]. The most frequent infections were complicated urinary tract infections caused by multiresistant microorganisms (29.8%), respiratory infections (23.2%), and intra-abdominal infections (19.9%). Remarkably, the mostly used antimicrobials were ertapenem (32.3%), ceftriaxone (25.2%), and piperacillin-tazobactam (11.4%). The median duration of OPAT was 8 days, with a median stay in at-home hospitalization of 10 days, avoiding conventional hospitalization in 34.8% of cases. In the two years of the study, 18,493 days of hospital admission were saved. The day-cost of admission to hospital was calculated as EUR 518/day, while HaH unit costs were calculated as EUR 98/day, which represents a savings of 81%. Considering the overall cost of the entire hospital stay, the savings were approximately 46% [56]. Another study conducted in the Barcelona area revealed that, after considering all pediatric OPAT episodes for a 2-year period, 2.62 beds/day were freed, with an economic benefit above EUR 1 million [57]. Moreover, unpublished data from 341 OPAT episodes in 2021 in our hospital determined an average cost of EUR 1770 per episode compared to the EUR 2657 for inpatient hospitalizations in the internal medicine service. The daily cost of inpatient hospitalization was EUR 232, while the OPAT daily cost was EUR 108, supposing an economic savings of 53%.

Finally, it must be noted that not all studies have shown economic benefits. Yong et al. compared patients undergoing OPAT with an equivalent group of hospitalized patients, considering both real costs and clinical results. They found that the cost of OPAT was reduced to 50% when compared to patients undergoing conventional hospitalization, but the overall costs were similar when clinical outcomes were considered [58].

## 3. Implementation of an OPAT Multidisciplinary Circuit

At Parc Taulí University Hospital, OPAT constitutes one of the main services of our HaH unit. Six years ago, as in most HaH units in Spain, OPAT was elaborated by nursing staff, which caused important limitations in terms of the microbiological stability of preparations, together with an additional workload. For this reason, the pharmacy department proposed to conduct OPAT preparation under sterile conditions in its laminar flow hoods, building a multidisciplinary team around the OPAT program with the essential participation of pharmacists. This team is mainly built up by a clinical pharmacist, a physician, and a nurse, all of them selected for their wide experience in infectious diseases and long-term self-administration treatments (Figure 1).

The main objective of the multidisciplinary circuit presented here is to improve the quality and safety of the OPAT program in our hospital. The circuit can be split into different stages: patient evaluation, medical prescription, clinical pharmaceutical validation, OPAT technical compounding and delivery, and self-administration (Figure 1). Next, we describe in detail the main features of each stage and compare our experience with published studies that have been analyzed through a non-systematic review.

### 3.1. Patient Evaluation

One of the main factors to guarantee the success of an OPAT program is the selection of adequate patients to be enrolled in the program. In our HaH unit, the evaluation team is built up of medical and nursing staff who are in charge of patient selection.

Mirón et al. established the OPAT protocols for the Spanish Society of Internal Medicine (SEMI) [12]. These protocols include the general selection criteria for the admission of patients in an OPAT program, which are detailed in Table 2 [1,12,59]. Selected patients should accomplish both the general criteria for HaH admission as well as the specific requirements for OPAT programs [60,61]. Among general HaH unit admission criteria, most authors agree on the importance of patients’ hemodynamic and clinical stability in order to achieve OPAT success. This stability is defined as apyrexia, normalized vital constants, and a stabilized or reasonably non-progressive infection course [62]. Additionally, the degree of treatment and administration-route-appropriate comprehension, together with cooperation between patients and families/caregivers is of utmost importance. In this sense, the success of an OPAT treatment depends on the ability for optimal self-administration (i.e., strength, abilities, and optimal seeing and hearing capacity as well as mental and physical fitness) [62].

High et al. gathered these criteria in a series of five questions (displayed in Table 3), aiming to define whether a patient is an optimal candidate for OPAT. When the answer to all questions is affirmative, OPAT becomes an optimal clinical option for the treatment of an infectious disease under circumstances of clinical stability [63].

### 3.2. Medical Prescription

As with any other type of prescription, when a physician decides to admit a patient into an OPAT program, they need to accurately prescribe both the optimal antimicrobial and the regimen (including the dosage and administration frequency). Additionally, differing from traditional prescriptions, OPAT regimens need to consider the final administration volume and the maximum concentration permitted as well as the flow rate and infusion time. The stability of the final product depends on these data and, in consequence, the choice of the optimal administration device for each patient and infection depends on the previously mentioned parameters [64]. Since several factors need to be considered, a joint evaluation and discussion by the whole multidisciplinary team is required.

As mentioned before, the administration device selection is also essential. More than 95% of OPAT prescriptions by our team are delivered with elastomeric pump devices after being elaborated under aseptic conditions. In our experience, these devices display several advantages, largely enabling OPAT self-administration and improving patients’ quality of life. First, elastomeric pumps can be self-administered by patients and stored at home for several days after receiving educational training by nursing staff. Next, the higher stability of antimicrobials prepared in these conditions permits the separation of HaH unit visits at a patient’s home, thus allowing them to include more patients in the program and enlarging the number of patients benefiting from the OPAT program. Finally, these devices allow the preparation of complex OPAT by including two or more antibiotics in a single device or using continuous perfusions in order to treat infections caused by multiresistant bacteria.

All these advantages have set elastomeric pumps as the most widely used OPAT devices, displacing electronic pumps, which present more complex functioning and are difficult to use by patients themselves at home. However, in some situations when specific antimicrobials lack technical data on stability in elastomeric devices, electronic pumps are still employed.

### 3.3. Clinical Pharmaceutical Validation: Involvement of Pharmacists in an OPAT Program

Clinical pharmacists must be part of the multidisciplinary teams in charge of OPAT programs. In fact, OPAT guidelines from the Infectious Diseases Society of America (IDSA) (Arlington, VA, USA) and the consensus document from the OPAT Project of Good Practice Recommendations Working Group in the UK advise that patient care should be conducted by a professional team with expertise in distinct fields, including a clinical pharmacist with experience in infectious disease management [65,66]. In this direction, a consensus document from SEHAD and the Spanish Society of Infectious Diseases and Clinical Microbiology (SEIMC) (Madrid, Spain) strongly recommends the incorporation of hospital pharmacists in OPAT teams, given their broad knowledge in antimicrobial stability, ADRs, drug interactions, pharmacokinetics monitoring, and the optimization of administration regimens (maximum doses and dilution volumes) [10]. The final goal is to improve and facilitate the workloads of healthcare professionals and caregivers as well as to enhance the safety of OPAT treatments.

Several publications have reported the advantages of a multidisciplinary OPAT team in approaching OPAT patients. Heinz et al. analyzed the impact of a multidisciplinary team (including a pharmacist) and the interventions performed to improve an OPAT program. Most of the actions taken by the team were related to safety issues (56%, such as dose adjustment to renal function) and the reduction of therapy complexity (41%, such as recommending continuous perfusion). Lastly, 29% of the interventions were associated with effectiveness (recommendations were made, for instance, on antimicrobial selection) [67]. Another study evaluated the adherence of HaH unit physicians to IDSA guideline recommendations [68]. After deciding to incorporate a supervision program led by clinical pharmacists, the compliance to IDSA guidelines was significantly improved. Further studies similar to this one are required to reassure the impact of pharmacists in preventing ADR appearance and hospital readmissions. In fact, clinical pharmacists exert an integral role in OPAT patient management, working in tight collaboration with infectious disease physicians [69]. This integral function includes highly relevant actions, such as participating in the selection of candidate patients and appropriate therapies, validating discharge prescriptions, closely controlling laboratory results, and periodically monitoring potential ADRs or venous access complications. These publications suggest that multidisciplinary teams constitute an optimal opportunity for hospital pharmacists to closely collaborate with physicians [68,69].

In summary, the positive results of including a clinical pharmacist in an OPAT multidisciplinary team can be divided into three categories: (i) intervention in antimicrobial stewardship programs (ASP) to optimize OPAT prescriptions, (ii) pharmacotherapeutic patient follow-up, and (iii) OPAT-specialized pharmacist technical counselling [70,71,72,73] (Figure 2). The pharmacist’s role in these categories is further developed in the following sections.

#### 3.3.1. Intervention in ASP to Optimize OPAT Prescriptions

First, pharmacists are responsible for validating that the antimicrobial prescription for OPAT is the appropriate one, given the patient’s clinical condition and the type of infection. Moreover, they have to assure that it will be administered in the most effective and safe manner, through the correct selection of the infusion device [10]. In general terms, the antimicrobial drug selection is not modified to follow in-hospital therapy at home, considering that it is the most optimal therapy for each patient. However, some antimicrobials do not possess the minimum stability requirements (such as cotrimoxazole and amoxicillin/clavulanate) to be part of an OPAT regimen. In these particular cases, hospital pharmacists are essential in defining the best alternative therapy. In addition, as often occurs in our own HaH unit, when the prescription involves a complex OPAT treatment (including two, or even three, antimicrobials with different administration regimens), pharmacists are responsible for ensuring that the whole therapy can be administered at home, maintaining the same effectiveness and safety standards as in an in-hospital setting.

Once the optimal antimicrobial(s) is(are) defined, the administration dosage and schedule need to be adjusted accordingly. Dose optimization is an essential part of ASP, and this role is generally assigned to clinical pharmacists, who make their specific recommendations to physicians. These adjustments are based on patient’s characteristics (such as weight, renal/liver function, and albuminemia) as well as the causative organism, the site of infection, and the antimicrobial’s pK/pD parameters [74,75]. During this review process, doses are often reduced due to several causes, with impaired renal function being the most common. However, certain disease contexts (endocarditis or bone and joint infections) require higher antimicrobial doses, similar to infections caused by highly resistant organisms, such as methicillin-resistant *Staphylococcus aureus* or multidrug-resistant *Pseudomonas aeruginosa*.

In parallel, the pK/pD profiles of the antimicrobial should be taken in consideration for an appropriate administration schedule. In the specific case of aminoglycosides, for instance, once-daily or extended-dosage regimens are preferred to traditional schedules (every 8 or 12 h) since they have demonstrated improved bacterial eradication concomitant with a lower incidence of ADRs (ototoxicity and nephrotoxicity) [75]. On the other hand, a recent study on ceftriaxone use in OPAT demonstrated no benefits of a single daily 4 g dose administration. Remarkably, the authors proved that spacing doses every 24 h reduced the time of exposure to an effective concentration and, as a consequence, the target exposure goal was only achieved for 50–60% of the time between doses. Therefore, this study suggests that a ceftriaxone regimen for *Enterococcus faecalis* should be maintained in a twice-daily fashion (2 g every 12 h) [76].

In the particular case of beta-lactam antibiotics, it has been reported that in-hospital patients benefit from the use of extended/prolonged or continuous infusions instead of traditional bolus administration, thus improving patient exposure to the antimicrobial drug, especially when using antipseudomonal beta-lactams [77]. This strategy was shown to improve clinical outcomes in critically ill patients and in infections caused by multiresistant organisms, where it decreases mortality, and we are convinced that outpatients in OPAT programs should also benefit from continuous beta-lactam infusions. Unfortunately, extended and continuous administration in an OPAT setting involves technical difficulties. For self-administration extended regimens (from 3 to 5 h), the availability of elastomeric devices is limited [64], as occurs in our HaH unit. In the case of continuous administration (24 h), for which adequate devices are available in our unit, the main handicap is the stability of the antimicrobial preparation, mainly in the spring–summer period, when stability is reduced due to higher temperatures. However, in recent years, an increasing number of publications support the use of continuous infusion pumps in OPAT programs [27,78,79]. When possible, OPAT teams should promote the use of extended or continuous infusions in the use of beta-lactams, specifically in those patients that can obtain a greater benefit. At this point, the role of clinical pharmacists in an OPAT team consists of overcoming the stability and technical issues mentioned above.

It is usually believed that OPAT can lengthen intravenous therapy, especially in antimicrobial treatment. In this sense, another important aspect of ASP in OPAT programs is to facilitate an intravenous-to-oral switch as soon as allowed by a clinical evolution of the infection and whenever an oral therapeutic alternative is possible [80]. For example, Dryden et al. reported that an ASP team (including the same professional profiles as our OPAT team) has a significant positive impact on the optimal use of antibiotics, such as increasing the switch from intravenous to oral treatment. This switch also involves positive effects in terms of reducing the risk of healthcare-associated catheter complications or infections [81]. Despite raising controversy on this point, it must be noted that recent OVIVA and POET (partial oral treatment of endocarditis) studies reported that the use of oral antibiotics at home in bone and joint infections and endocarditis (among the most common infection types in OPAT programs) displayed no inferiority when compared to maintaining intravenous in-hospital therapy [82,83]. In both studies, patients received a maximum of two weeks of intravenous therapy during their hospital admission before being randomized into two groups: either completing intravenous treatment or switching to the oral route upon discharge. Presently, the oral switch is not widely implemented because stronger evidence is still required. Instead, solid evidence over the years is set on the effective and safe use of OPAT for endocarditis treatment [84].

#### 3.3.2. Pharmacotherapeutic Patient Follow-Up

Outpatients in OPAT programs deserve the same degree of pharmaceutical attention as if they were admitted to hospital and undergoing an infectious process. Therefore, once an OPAT prescription is fully validated, clinical pharmacists should conduct patient management following the same criteria as for in-hospital patients. Initially, upon a patient’s inclusion in an OPAT program, previous comorbidities should be considered and evaluated in order to complete an optimal pharmacotherapeutic review and a proper medication conciliation. Since the healthcare level of patients is modified when they are discharged and initiate OPAT at home, this change is an opportunity to detect potentially inappropriate prescriptions in their chronic treatments [85]. This is especially important in geriatric patients, whose admissions in HaH units are constantly increasing.

In this sense, it is also essential to exhaustively corroborate potential allergies to distinct antimicrobial families. Patients are commonly reported to be allergic to specific antimicrobial drugs [86]. However, mainly in the case of beta-lactam allergies, most of them are not real allergies. In fact, some patients labelled (or self-labelled) as allergic to penicillin are not allergic when properly tested or re-challenged. These “fake” labels lead physicians either to prescribe second-line antimicrobials (therefore a suboptimal drug choice) [86] or to treat patients with broad-spectrum antibiotics, thus increasing the risk of antimicrobial resistance and ADRs [87]. In some cases, patients are discharged to OPAT programs since they fulfill all inclusion criteria, despite a lack of results for their microbiological tests. In these situations, the whole multidisciplinary team needs to be aware of the results, given that ASP interventions may be required upon the availability of microbiology data [70,88].

Therapeutic drug monitoring (TDM) needs to be carefully conducted during OPAT as it occurs with in-hospital patients under antimicrobial therapy. Douiyeb et al. retrospectively studied the risk factors for OPAT patient readmission. Remarkably, discharge with vancomycin or aminoglycosides was found to be an independent risk factor for the readmission of these patients since only half of them were properly monitored according to the IDSA guidelines [89]. Consequently, adequate drug monitoring is recommended to prevent readmissions and complications. The most common antimicrobials included in TDM are vancomycin and aminoglycosides. It would be useful to monitor the levels of other antibiotic families, but it is known that resources and funding are limited in many institutions. In these cases, publicly available data define the therapeutic ranges of antimicrobials, a valid tool for treatment optimization and ADR minimization [90]. In our HaH unit, we maintain TDM on further patient treatments beyond antimicrobials, such as immunosuppressive or antiepileptic drugs.

During OPAT, intravenous treatments should not be the only focus of attention for pharmacists, since they are often complemented by the oral intake of further antimicrobials, increasing the risk of suffering ADRs and/or drug interactions [13]. This issue acquires more relevance in elderly patients with polypharmacy that, as mentioned above, are progressively increasing in number. In fact, in our OPAT program, more than 60% of patients in the last three years were >65 years, including 15% above 90 years of age. A recent study demonstrated the safety of OPAT in a nonagenarian cohort, where higher mortality was found in this group, although it was demonstrated to be irrespective of OPAT-derived complications [91]. Apart from ADRs derived from drug-to-drug interactions, in an OPAT context clinical pharmacists have to closely monitor the potential appearance of ADRs related to the prolonged use of some antimicrobials. For instance, high levels of voriconazole induce hepatotoxicity, while toxicities associated with the prolonged use of ganciclovir include pancytopenia, neurotoxicity, and nephrotoxicity, all of them described as dose-independent effects [92]. In addition, a recent systematic review and meta-analysis concluded that daptomycin administration in OPAT patients under treatment with statins significantly increased the incidence of rhabdomyolysis caused by creatinine phosphokinase level elevation [93].

In this context, clinical pharmacists hold an ideal position within OPAT teams to guarantee proper clinical management from the initial antimicrobial drug selection and throughout every stage of the treatment, aiming to provide effective and safe OPAT healthcare attention.

#### 3.3.3. OPAT-Specialized Pharmacist Technical Counselling

Beyond the validation process directly related with antibiotic selection and prescription, pharmacists in HaH units confer an essential value to multidisciplinary teams, given their knowledge in the technical aspects related to OPAT preparation [72]. At this point, several factors such as antimicrobial stability, solvent choice, the final antimicrobial concentration, compatibility between the drugs and materials used, the type of infusion device, and storage conditions should be considered [64].

Antimicrobial stability is an essential factor that determines the organization of HaH units and, specially, the workloads of pharmacists and compounding technicians in OPAT teams. In fact, the frequency of visits by our HaH unit at patients’ homes is determined by drug stability (unless a patient requires special medical assistance), as it occurs with the scheduling of workflows at the pharmacy department. Information on antimicrobial stability in elastomeric pumps can be obtained through distinct sources: databases from elastomeric pump manufacturers (with access usually restricted to customers), the Stabilis^®^ database, systematic published reviews [94,95,96], official guidelines or consensus documents [10,65], or specific publications about a single antimicrobial or family [97,98]. Sometimes, a lack of stability data or short stability time are the main factors that prevent a patient from entering an OPAT program or cause the replacement of an initially prescribed antimicrobial.

Moreover, the weather conditions of the area also influence antimicrobial stability. In fact, most of the available technical data are obtained at 25 °C [95], but this is not representative of some periods of the year in the area of our HaH unit, where this is commonly exceeded in spring and summer. This constitutes a problem for extended infusions, although there are publicly available data supporting the use of these perfusions in higher temperature conditions [99,100,101]. In our hospital, when the OPAT multidisciplinary circuit was built, a thorough research on the stability times of the most frequently used antibiotics was carried out by specialized pharmacists, using bibliography data for reference. However, this is a dynamic process in which novel stability studies are constantly published. Consequently, our stability data are periodically updated when required. Unfortunately, there are insufficient data on the stability of some antimicrobials (e.g., cotrimoxazole or anidulafungin) [95,102], and they are not yet included in our OPAT program.

The selection of the most appropriate solvent depends on factors from the antimicrobial itself and/or the patient. Regarding drug-associated factors, we should select the solvent with physicochemical compatibility that confers the longer stability to OPAT. When checking patient-related factors, in some cases the selection depends on analytical parameters such as electrolyte disorders or hyperglycemia. In our OPAT program, according to available stability studies [102], the main solvents used are a 0.9% sodium chloride solution and a 5% glucose solution. It must be noted that some antimicrobials are only stable in a single solvent. For instance, liposomal amphotericin B can only be diluted in 5% glucose, while daptomycin is only compatible with 0.9% sodium chloride [102,103,104]. For this reason, it is always recommended to corroborate an antimicrobial’s solubility in the summary of product characteristics. 

The final concentration of antimicrobials in OPAT preparations is another factor to be considered. In fact, some drugs are more stable at lower concentrations and, therefore, lose stability when they are highly concentrated, with meropenem being a clear example of this inconvenience [105,106,107]. For this reason, when an OPAT dose needs to be modified, it is important to verify whether the stability and expiration period will be maintained at the new concentration. In addition, the osmolarity of the final solution should always be kept within the plasma range to avoid potential damages in venous access [108]. In parallel, increasing the concentrations of some antimicrobials leads to excessive viscosity. When administration is performed with elastomeric pumps, this increased viscosity may alter the flow infusion rate. In our initial experience, piperacillin/tazobactam elastomeric pumps could not be fully emptied in 24 h continuous infusions due to viscosity issues. This problem is caused by the high salt content of the formulation [109]. Since the dose could not be reduced and we lack elastomeric pumps with higher infusion rates for the established 24 h period, we were forced to reduce the total volume of the solution, thus establishing the optimal infusion time.

In OPAT preparation, it is also important to corroborate the compatibility not only between the antimicrobials used and solvents but also with the material of the elastomeric devices used. This consideration is even more relevant when elaborating complex OPAT, including more than one antimicrobial together. In this case, we need to confirm that all antimicrobials can be jointly included in a unique device. This is the case of an ampicillin and ceftriaxone combination, which is prescribed for the treatment of endocarditis by *Enterococcus faecalis*. A recent study demonstrated that both antimicrobials can be administered in a complex OPAT combination since the stability of both drugs (containing a minimum of 90% of the initial included amount) is maintained at two different temperatures (25 °C and 30 °C) [110]. We recommend that, in similar situations, the clinical pharmacist should corroborate the compatibility of the whole OPAT treatment before its elaboration.

In our opinion, when implementing an OPAT program in a HaH unit, it is not necessary to acquire all types of commercial elastomeric pumps. In fact, 3–4 types of devices, covering all potential needs of volumes and infusion rates, is sufficient. Most antimicrobials can be administered in 30–45 min in 100 mL pumps, which have an appropriate infusion rate for these preparations. In parallel, some antimicrobials require a larger volume for stability and safety issues, and in this case, 250 mL pumps can be employed, while others must be administered in 1 h devices (such as ceftaroline, ganciclovir, or imipenem). Last but not least, continuous perfusion devices are required for beta-lactams, with a minimum administration time of 4–5 h [77]. In order to reduce the types of devices employed, our HaH unit only uses 24 h devices for these cases since we have no elastomeric pumps available for intermediate times. The particular organization of our HaH unit is an example of how an adequate strategic selection of the devices used for an OPAT program is essential for the optimal management of stocks and employed resources.

Once the antimicrobial is diluted in the final device for administration, the storage conditions, such as the temperature conservation range or light protection requirements, should be optimal. For instance, ceftolozane/tazobactam should never be in contact with direct light [111], and almost all antimicrobials need to be refrigerated during storage, with the exception of cefepime and micafungin. In both cases, low temperatures negatively alter drug stability. When an OPAT treatment contains thermolabile antimicrobials (i.e., most of the time), an appropriate storage space for elastomeric pumps is necessary, both in the pharmacy department and in patients’ homes. The cold chain needs to be strictly maintained from the initial preparation to the final administration, including during transportation between the hospital and the final destination. This chain should only be broken for OPAT tempering 30 min before self-administration. This is an important point to be raised during patient education by nursing staff.

Voumard et al. analyzed the stability of five different antimicrobials administered in continuous perfusion in elastomeric devices. Remarkably, the study demonstrated that the OPAT temperature can increase up to 30 °C at night when the device is attached to the patient’s body under bed linen. Instead, when the OPAT device is located beside of the bed (e.g., on a table), the temperature does not exceed 25 °C. This is not a minor issue since most of the stability data of antimicrobials are obtained at temperatures below 25 °C. Therefore, an inadequate location for the device during night administration can drastically alter OPAT stability, as in the case of a cloxacillin preparation [112]. Since global temperature increase is a real fact, these situations are becoming more common and, consequently, more real-life studies are required to collect more data on OPAT storage conditions and stability. The Perks et al. publication analyzing the influence of warm climate environments on extended perfusions is an example of this type of study [99].

In conclusion, the involvement of the pharmacy department in OPAT teams should contribute to the close follow-up of patient evolution and OPAT results in terms of safety and effectiveness and should not be limited to assistance in prescription and compounding. Detailed lists of quality indicators to evaluate the robustness of OPAT programs have been published [19,113] and constitute an ideal entry point for novel OPAT teams aiming to improve the quality of their services.

### 3.4. OPAT Technical Compounding and Delivery

After the pharmaceutical validation, the pharmacy department exerts a fundamental role on the next step of the process, which is the technical compounding of an OPAT treatment before it can be delivered to outpatients.

In most of the cases, antimicrobials need to be manipulated from the commercial presentation to the final OPAT preparation. Vial manipulation (reconstitution or dilution) involves a potential contamination risk or the alteration of the physicochemical properties of the product. Therefore, the most accurate compounding methodology in an optimal work environment is required [114]. In this sense and given the sterility requirements for intravenous administration, OPAT preparations are included in the pharmacy aseptic preparation services within the compounding area. Technical personnel with specific training on injectable medicine compounding should be responsible for OPAT preparation. Importantly, this process should be conducted in clean rooms containing laminar flow hoods, with the final goal of minimizing the possible microbiological contamination. The clean room environment is subjected to a periodic control of parameters such as air particles, temperature, light, or air flow [115,116].

In fact, there is strong evidence about the benefits of preparing injectable medicines in the controlled environment of clean rooms. In a 2019 meta-analysis, the contamination rate of parenteral medication in hospitals prepared either in a pharmacy department or in clinical environments was compared. A total of 13 studies published between 2000 and 2018 were included, reporting that the contamination rate of sterile medication prepared in clinical environments was 7.47%, whereas doses prepared in pharmacy department clean rooms only presented a 0.08% contamination rate. The study also indicates that parenteral medications should be compounded by trained pharmacy staff [117]. Apart from increasing patient safety by reducing microbiological contamination, the further advantages of injectable medicines preparation in our clean rooms are: (i) to guarantee their physicochemical stability, storage conditions, and expiration dates; (ii) to prevent medication errors; (iii) to achieve a better integration of pharmacists into multidisciplinary teams; (iv) to reduce variability in preparations due to standardized protocols; and (v) to optimize material and human resources [102,118].

For all these reasons, healthcare institutions recommend the creation of centralized drug compounding units where all in-hospital injectable medicines should be prepared. Since OPAT is considered a parenteral medication, unlike the traditional preparation in clinical units by nursing staff [118,119], it should also be included within the services of these centralized units [102,120]. These units are responsible for fulfilling the standard regulation and procedures related to sterile medication compounding, therefore warranting the quality of all prepared OPAT products [72,115,116]. As an example, in our own HaH unit, the pharmacy department took over the responsibility for OPAT preparation by experienced personnel in our clean rooms. This change led to the assignment of longer expiration periods to antimicrobials, spacing the delivery of elastomeric pumps, and allowing more flexibility in home visits by physicians and nursing staff.

#### Compounding and Delivery Procedure in our OPAT Program

Initially, the pharmacist responsible for OPAT validation organizes a daily schedule for all elastomeric pumps to be prepared by technical staff. The following information should be included: patient identification, corresponding antimicrobial and prescribed dose, elastomeric units to be elaborated, and the number of units to be delivered. High-stability antimicrobials permit long-term storage, conferring higher adaptability to the planning of OPAT compounding [72].

Following this schedule, the elastomeric devices are prepared, maintaining the traceability of the product during the whole process. Therefore, the technical staff keeps a record of all data related to production for each OPAT product. The technical production document sheet includes the date of preparation, code number assignment, antimicrobial vials and elastomeric pumps used (including batch number and expiration date), amount prepared, beyond use date of the final product, storage conditions, and manufacturer’s identification [115]. Moreover, each elastomeric pump is individually labelled in order to unambiguously identify both the patient and the antimicrobial regimen, providing the essential information for correct self-administration [72]. At this point, the whole procedure and the final product are validated by the pharmacist responsible for the compounding area. Finally, elastomeric pumps are packaged in appropriate conditions for cold chain maintenance (i.e., with cold accumulators), and they are ready to be delivered in a patient-individualized fashion.

Although OPAT preparation is scheduled on a daily basis, the delivery calendar is organized in weekly terms. This weekly plan, agreed on by the HaH unit and pharmacy department, is based on a tight equilibrium between the stability of the antimicrobials and the workloads of the healthcare professionals involved. For instance, antimicrobials with high stability, such as ceftriaxone and caspofungin, are delivered twice weekly, whereas drugs with shorter stability periods are delivered with a frequency from three times a week to once daily. However, this schedule is flexible and is always adapted to the clinical requirements since patient healthcare is our main priority.

### 3.5. Self-Administration

The last step of our OPAT multidisciplinary circuit consists of the administration to selected patients. OPAT represents approximately 40% of the nursing activity of our HaH unit, and it has differential requirements compared to the other activities of home care. For instance, specific devices are needed for treatment administration, and the nursing staff in charge of the program should be specialized in this type of administration, specifically in the context of infectious diseases. Moreover, the willingness and active involvement of patients and caregivers is probably the most essential aspect for OPAT program success [10].

Every HaH unit designs its healthcare practice according to the needs of the reference hospital, the geographical area of influence, the complexity level of patients, and the availability of material and human resources. In our case, more than ten years ago the HaH unit decided to implement OPAT self-administration with elastomeric devices as the preferential therapeutic option [35]. Remarkably, OPAT self-administration requires us to invest time on patient education for the proper use of the devices, which has to be adapted and targeted for each case, followed by a continuous evaluation during the healthcare process. The goal is to provide patients and caregivers with the appropriate knowledge and skills, favoring self-care and their involvement in decision making. In fact, it was reported that an OPAT program under appropriate nursing staff surveillance decreases the rate of in-hospital readmissions [121].

With this purpose, the main healthcare actions of nursing staff in an OPAT program are summarized in: (i) establishing a patient–nurse relationship based on trust and fluent communication by enabling proper communication channels with the patient; (ii) achieving patient adherence to treatment through a targeted educational program, including the elimination of generated residues; (iii) guiding patients and caregivers to the optimal use and care of venous access; (iv) scheduling periodic communication with patients, either by phone contact or face-to-face visits; (v) ensuring the appropriate delivery of OPAT devices and the adequate storage at patients’ homes; and (vi) performing analytical controls and microbiology cultures when necessary [12,59].

From all these actions, we would like to emphasize the important role of nursing staff in the selection and care of vascular catheters and counselling on storage conditions to prevent OPAT degradation.

#### 3.5.1. Selection of Vascular Catheter and Catheter Care

An adequate selection of venous access determines correct OPAT administration and prevents potentially serious complications [122]. The main factors that influence the selection of an optimal catheter are the treatment length, the patient’s venous access, and the antimicrobial’s characteristics [59]. Depending on the treatment duration, our HaH unit mainly employs three types of catheters:PICC: Used for long-term OPAT (>3–4 weeks) and allows multidose daily administration as well as bitherapy.Midline catheters: Employed for mid-term OPAT (between 2 and 3 weeks). Mainly used for once- or twice-daily OPAT administration. Our team has also successfully employed midline catheters for OPAT regimens with frequencies of every 6–8 h.Peripheral venous catheter: Used for short-term OPAT (less than 10 days of treatment) and only recommended in once-daily regimens.

In specific situations, for already-catheterized patients (e.g., with a tunneled central venous catheter or a subcutaneous reservoir), it is possible to administer OPAT through the same catheters.

Once a catheter is placed and the patient is part of an OPAT program, the nursing staff has to evaluate and check the correct insertion of the catheter every 24–48 h, combined with specific antithrombotic care. When required, healing and dressing actions on the catheter insertion point are executed (e.g., bandage replacement and disinfection of the area with 70% ethanol). During each home visit, the nursing staff makes a visual check of the catheter insertion point appearance and corroborates venous access permeability. In cases of local pain, erythema, phlebitis, or catheter obstruction, the nursing staff advise the OPAT team on a possible catheter replacement.

In parallel, patients and caregivers have to be aware of important considerations regarding the self-care of catheter insertion. The nursing staff should emphasize the importance of optimal hand hygiene before catheter manipulation, combined with sterilization of the connection point between the catheter and the OPAT device with an ethanol solution as well as keeping the catheter and dressing permanently dry and protected from external elements. The educational process should contain how to instruct the patients/caregivers with tips to identify catheter-related complications. The most frequent complications are catheter obstruction or perfusion slowing, insertion point leakage or bleeding, accidental catheter withdrawal, phlebitis signs, or fever. In any of these situations, the patient should be advised to immediately contact the OPAT healthcare team [123,124].

It must be noted that, mainly in geriatric patients, it can be difficult to find an optimal venous access for catheter insertion. For these situations, outpatient subcutaneous antimicrobial therapy (OSCAT) has become a widely accepted alternative [125] that is also accepted in our HaH unit. However, this route of administration is not yet approved for any antimicrobial in the summary of product characteristics, although some publications support OSCAT, mainly for antimicrobials with longer half-lives, such as ceftriaxone and ertapenem [125].

#### 3.5.2. Counselling on OPAT Storage Conditions

When initiating OPAT treatment, the nursing staff advise the patients and caregivers on how to properly store elastomeric devices at home, with the main goal to prevent temperature increase. With this goal, OPAT devices should be hung and never placed under clothes nor attached to the body. Moreover, patients should go outside during mornings with cooler temperatures, especially in summer, and protect the device from direct sunlight. At night, the elastomeric pump should be left outside the bed and never kept under the bed linen to prevent high temperatures that may degrade the antimicrobials [64].

In our opinion, beyond OPAT effectiveness, patients’ degree of satisfaction with the whole OPAT team is an essential aspect to pursue the continuous improvement of our multidisciplinary team activity. At this point, the role of nursing staff is pivotal since they are in direct contact with both patients and caregivers and can, therefore, detect potential complications through their constant presence and support them throughout the process.

## 4. Managing Pharmacy Department in an OPAT Multidisciplinary Circuit

In healthcare terms, innovation can be implemented at different management levels (macro-, meso-, or micromanagement), while the object of innovative efforts can be either a product, a process, or an organization method. New models of healthcare that involve multidisciplinary teams are explained by meso- and micromanagement under the perspective of innovative processes. Frequently, these innovative processes are related to changes in the way of facing challenges as a team, and the results of those challenges appear as a collaborative patient approach [126].

The multidisciplinary circuit presented here originated from the emerging need, detected back in 2013, that antimicrobial device manipulation should be improved for the higher quality and safety of the OPAT program [96,127]. From the point of view of management, the pharmacy department struggled with the dedication to achieve the proper resources to improve these procedures. As it is known, it is necessary to identify which indicators of activity, structure, and process would be useful to monitor or evaluate any new circuit, and the result of those indicators was strategic for building OPAT necessities related to time and human resources, which were addressed to provide the team with all the required components. It was finally in 2016 when the expected resources were obtained, with the incorporation of a clinical pharmacist and a sterile compounding technician dedicated to ensuring the quality of OPAT preparations, leading to the birth of the complete multidisciplinary OPAT team.

Since each OPAT team presents its unique characteristics and needs, pharmacy department resources for OPAT preparation should be adapted to each situation. Not all realities are the same, and not all workloads are comparable. Therefore, it would be useful to use external benchmarking information to understand data characterization, the services offered, automation, workflows, and workloads before designing workforce planning [128]. In this sense, developed countries have proposed indicators that can be used to benchmark OPAT programs. For instance, it is considered that one OPAT pharmacist full-time equivalent (FTE) is required for every 45–70 OPAT patients [129]. However, apart from human resources, in terms of pharmacists and technicians [128], the implementation of an OPAT program as described here requires optimal infrastructure, including certified clean rooms and the availability of infusion devices. Unfortunately, these requirements are not available worldwide, leading to a low implementation of OPAT programs throughout the least developed countries, such as Asian countries, where only 3% of hospitals present a comprehensive program with specialist oversight [130]. Therefore, benchmarking indicators are a useful tool to determine consensus high-quality standards and, beyond peer comparison, can be used by each organization to define its own strategies in order to achieve these quality standards.

In summary, OPAT is a clear example of healthcare innovation through a multidisciplinary patients’ approach. Some years ago, there was no clear idea about which specific contribution a clinical pharmacist could bring to an OPAT team [67,131,132]. Presently, it has been demonstrated that the pharmacy department can exert a key function in multidisciplinary OPAT circuits, as has been widely described throughout this review. Consequently, clinical pharmacists emerge as new essential figures in HaH units in charge of OPAT programs.

## Figures and Tables

**Figure 1 antibiotics-11-01124-f001:**
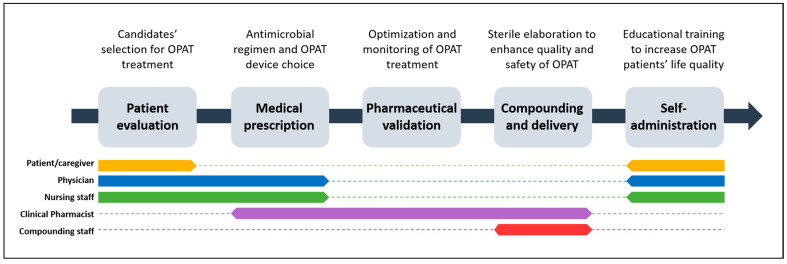
OPAT multidisciplinary circuit at Parc Taulí University Hospital. The central diagram indicates the five stages of the circuit. The headings above summarize the main contribution of each stage to the OPAT program. The bottom colored dotted lines individually represent patients/caregivers, all members of OPAT team, and the compounding staff. Thicker lines below a specific stage indicate a major contribution to the corresponding stage.

**Figure 2 antibiotics-11-01124-f002:**
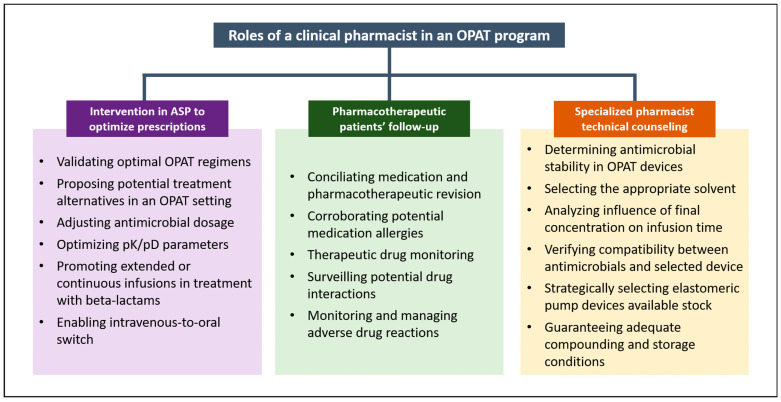
Main roles of clinical pharmacists in the OPAT multidisciplinary team presented in this review. The roles are divided into three main categories: intervention in ASP programs to optimize prescriptions, pharmacotherapeutic patient follow-up, and specialized pharmacist technical counseling.

**Table 1 antibiotics-11-01124-t001:** Decalogue defining OPAT care model by SEHAD [8].

Spanish Society of Hospital at Home OPAT Decalogue
OPAT must be understood as a clinical therapeutic procedure that requires a well-organized care structure for its performance. HaH ^1^ is the ideal resource for the implementation of an OPAT program. Professionals in OPAT programs must acquire a set of specific skills and knowledge about clinical practice, pharmacology, and microbiology as well as the use of catheters and drug infusion devices. Professionals in OPAT programs must have communication and empathy skills to generate trust and satisfaction. HaH units must establish a professional/patient ratio that guarantees safety and quality care. OPAT must follow the recommendations of scientific societies and experts on the rational use of antibiotics. Complex OPAT is defined by a high technical and preparation complexity, either due to the type of germ or infection, and aims to improve the pK/pD ^2^ parameters of the antimicrobial to reduce its ADR ^3^ or to optimize the antimicrobial stewardship programs. Complex OPAT is a multidisciplinary procedure involving, at least, nurses with experience in OPAT, physicians with advanced knowledge in infectious diseases, and hospital pharmacists. Self-administered OPAT is a safe procedure when supervised by professionals and performed by patients or caregivers with the ability to follow the given instructions. OPAT is an efficient, safe, and cost-effective procedure when it is performed under controlled conditions by qualified professionals with the optimal resources.

^1^ Hospital at Home; ^2^ Pharmacokinetic/Pharmacodynamic; ^3^ Adverse Drug Reactions.

**Table 2 antibiotics-11-01124-t002:** General criteria for HaH and OPAT admission. Adapted with permission from [12].

**General Criteria for Patient Admission in an HaH Unit**
Willingness of both the patient and family/caregivers (with informed consent); Availability of optimal communication by phone; Residence within the region of the healthcare system range; Ideal hygienic and sociofamiliar conditions; Absence of acute decompensation of a psychiatric pathology and active use of alcohol and abuse drugs; Clinical and hemodynamic stability of the patient.
**Specific Admission Criteria for OPAT Programs**
Requirement for intravenous administration of antimicrobial, given the lack of available and/or recommended alternative options; Availability of venous access, according to the type of antibiotic and the length of treatment; Administration of the first dose in a hospital setting, with the exception of specifically selected patients; Deep understanding of the proposed treatment and high cooperation between patient and family/caregivers.

**Table 3 antibiotics-11-01124-t003:** Questionnaire to evaluate the adequacy of patient enrolment in an OPAT program. Adapted from [63].

Is the intravenous treatment necessary, and are there no further therapeutic options with an equivalent effectiveness through the oral route? Is the care site (patient’s home) appropriate? Is the patient under a situation of clinical stability? Can either the patient or family/caregivers correctly comprehend and execute the treatment administration? Is effective communication guaranteed?

## Data Availability

Not applicable.

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
