# Peer review of "Successful Integration of Clinical Pharmacists in an OPAT Program: A Real-Life Multidisciplinary Circuit"

_antibiotics, 2022, doi:10.3390/antibiotics11081124_

Round 1

Reviewer 1 Report

General comment

·         Write the full form of ‘OVIVA’.

 Specific comments

·        Lines 63-65, lines 279-283, lines 416-418, lines 613-614 and lines 839-841: Rewrite the sentences clearly.

·         Lines 65-68: Cite the references to those ‘other studies’. Also, present sufficient background information to support your opinion as mentioned in the sentence.

·         Lines 206-208, lines 220-222, lines 520-521, and lines 634-635: Cite the reference to support the data or statement provided in the sentences.

·         Lines 218-220: You’ve mentioned ‘Further studies …’ but have cited only one reference.

·    Lines 475-478 and lines 478-481, lines 501-503, lines 545-546, lines 584-587, lines 593-595, lines 600-601, lines 616-617, lines 618-621, lines 633-634, lines 641-645, lines 700-706, lines 713-716, lines 750-752, lines 760-761, lines 785-786, lines 816-817, and lines 831-833: I wonder how you compared other findings with yours in these sentences since this was a review manuscript.

·         Lines 570-571: No need of definition of antimicrobial stability

·         Lines 838-839: Sentence not needed

 References

·         References 3-5, 8, 10-12, 20, 59, 79, 102, 116, 119, 122, 126: Rewrite names of the authors properly.

Author Response

Reviewer 1

General comment

  • Write the full form of ‘OVIVA’.

As suggested by the reviewer, we have included the full name of OVIVA trial, in line 60 of the new version (with Track Changes function activated). To facilitate comprehension of the text, we have also included the full form of “POET” trial, in line 514 of the new version.

 Specific comments

  • Lines 63-65, lines 279-283, lines 416-418, lines 613-614 and lines 839-841: Rewrite the sentences clearly.

We thank the reviewer for their comment and agree that those sentences were not written clearly enough. We have reformulated these sentences to improve reading, as it is reflected in the new version of the manuscript.

  • Lines 65-68: Cite the references to those ‘other studies’. Also, present sufficient background information to support your opinion as mentioned in the sentence.

In these lines, we did not refer to “other studies” as the reviewer mentions, but to “other authors”, in a clear reference to SEHAD decalogue. However, to make it clearer, we have incorporated reference [8] in the text. Moreover, we have rephrased the last sentence of the paragraph, reinforcing that we present our opinion about the concept of “complex OPAT”, in agreement with SEHAD.

  • Lines 206-208, lines 220-222, lines 520-521, and lines 634-635: Cite the reference to support the data or statement provided in the sentences.

In lines 206-208, we have added reference [41], since data are included in this study. However, we believe it was already clear because it was cited just after the next sentence.

Lines 220-222: This sentence describes our own experience and, since it is not published yet, we cannot provide any reference. Therefore, we have removed the part of the sentence where it refers to safety issues.

Lines 520-521: Reference [86] has been included in the new version.

Lines 634-635: The sentence has been reformulated, emphasizing that it is an opinion/recommendation from the authors, based on our experience.

  • Lines 218-220: You’ve mentioned ‘Further studies …’ but have cited only one reference.

Thanks to the reviewer for the comment. We have now corrected the text and, instead of “Further studies”, the sentence begins referring to the study itself (Seo et al. 2020)

  •   Lines 475-478 and lines 478-481, lines 501-503, lines 545-546, lines 584-587, lines 593-595, lines 600-601, lines 616-617, lines 618-621, lines 633-634, lines 641-645, lines 700-706, lines 713-716, lines 750-752, lines 760-761, lines 785-786, lines 816-817, and lines 831-833: I wonder how you compared other findings with yours in these sentences since this was a review manuscript.

We agree with the reviewer that, in many of these sentences, we make reference to our HaH unit or OPAT program and we understand that it is not the usual content in a Review manuscript. However, when we were invited by the Editors to make a contribution to this Special Issue, we proposed to write a non-systematic review about OPAT circuits, and to add a particular value to it, by describing our own experience. The Editors accepted our proposal and, therefore, the manuscript is structured like this.

Nevertheless, to advise potential readers about the content, we have added the following sentence: “Next, we describe in detail the main features of each stage, and compare our experience with published studies that have been analysed through a non-systematic review”. The new sentence is located in lines 344-345 of the new version with Track Changes activated, preceding all sections where we compare previous literature with our experience.

In addition, we have modified some of the sentences mentioned by the reviewer, in order to reduce the number of unpublished self-references and, in others, we have incorporated citations with the information provided. For example, in lines 501-503, we have removed “in our daily practice” and in lines 584-587, we have added reference [95] to support our statements. The remaining modifications in these specific lines can be found in the new submitted version.

  • Lines 570-571: No need of definition of antimicrobial stability

We thank the reviewer for his/her comment, and we have removed this definition from the text.

  • Lines 838-839: Sentence not needed

Again, thanks for this comment. We have removed the sentence accordingly.

 References

  • References 3-5, 8, 10-12, 20, 59, 79, 102, 116, 119, 122, 126: Rewrite names of the authors properly.

Thanks for deeply checking the references of the manuscript. We have corroborated all the references mentioned by the reviewer and have addressed the ones containing mistakes in the name of the authors (references 4, 8, 12, 20 and 116). Regarding the other citations, the names of the authors appear exactly as in the original papers. We are not sure, but maybe the problem is the lack of dash between authors’ surnames, since most of the references are from Spanish authors in Spanish journals. However, we understand that names of the authors have to appear as in the original source.

Reviewer 2 Report

The current manuscript presents a quite interesting view on the Outpatient Parenteral Antimicrobial Therapy (OPAT) programs, and the role that the clinical pharmacist can have in this sort of programs. It explores the issue deeply, and provides an analytical view of both its advantages, disadvantages and general applicability. It is also quite well written.

The only change that I ask for is the following: in section 4, it says “Since each OPAT team presents its unique characteristics and needs, Pharmacy Department resources for OPAT preparation should be adapted to each situation. Not all realities are the same, and not all workloads are comparable. Therefore, it would be useful 860 to use external benchmarking information to understand data characterization, services offered, automation, workflows and workloads before designing workforce planning”; could you please expand this paragraph to include some more specific strategies that could be applied in order to adapt this sort of approach to different realities? For example, differences between developed and underdeveloped countries, richer or poorer economies, etc.?

I recommend the acceptance of this manuscript after this minor revision.

Author Response

Reviewer 2

The current manuscript presents a quite interesting view on the Outpatient Parenteral Antimicrobial Therapy (OPAT) programs, and the role that the clinical pharmacist can have in this sort of programs. It explores the issue deeply, and provides an analytical view of both its advantages, disadvantages and general applicability. It is also quite well written.

We thank Reviewer 2 for his/her positive comments about the manuscript. We are glad to read that he/she considers our manuscript interesting and valuable.

The only change that I ask for is the following: in section 4, it says “Since each OPAT team presents its unique characteristics and needs, Pharmacy Department resources for OPAT preparation should be adapted to each situation. Not all realities are the same, and not all workloads are comparable. Therefore, it would be useful 860 to use external benchmarking information to understand data characterization, services offered, automation, workflows and workloads before designing workforce planning”; could you please expand this paragraph to include some more specific strategies that could be applied in order to adapt this sort of approach to different realities? For example, differences between developed and underdeveloped countries, richer or poorer economies, etc.?

We agree that this paragraph could be expanded and believe that his/her suggestions are of great interest to potential readers. It is obvious that not all health systems are at similar levels of development and patients in some underdeveloped countries may not be candidate to OPAT programs as it occurs in our environment. Therefore, we have added more considerations on this topic and have included two new citations, about benchmarking strategies and comparing our situation with that of underdeveloped countries.

I recommend the acceptance of this manuscript after this minor revision.